# Cork, a Natural Choice to Wine?

**DOI:** 10.3390/foods11172638

**Published:** 2022-08-30

**Authors:** Joana Azevedo, Paulo Lopes, Nuno Mateus, Victor de Freitas

**Affiliations:** 1LAQV—REQUIMTE-Laboratório Associado para a Química Verde, Faculdade de Ciências da Universidade do Porto, Rua do Campo Alegre 687, 4169-007 Porto, Portugal; 2Amorim Cork S.A., Rua dos Corticeiros 830, 4536-904 Santa Maria de Lamas, Portugal

**Keywords:** wine, cork stoppers, interactions, polyphenols, corklins

## Abstract

This review presents the most recent data on the state-of-the-art of the main compounds present in cork, their interaction with wine, and the impact that natural stoppers may have on wines’ physical-chemical and sensory properties. According to the recent scientific literature, the chemical composition of cork and the scientific relevance of the compounds extract from cork to wine over time are reviewed. Furthermore, the effect of cork compounds transfer into wines during post-bottling is also discussed, as well as their impact on the organoleptic (colour and taste) of wines. This knowledge is essential for the decision-making process undertaken by wine producers to select the stopper most suitable for their wines. In addition, sustainability is also a topic addressed since it is a natural product that generates some waste as well as the way in which this industry is adapting to the closure of the waste cycle.

## 1. Introduction

The cork oak tree is very important for Portugal, and in 2011 the Portuguese Parliament declared this tree Portugal’s National Tree, reinforcing the protection acquired by law since the 13th century. Historically, cork was first used by Greeks, Romans, and Egyptians as a closure for amphoras [1]. In the early 1600s, Dom Pérignon innovated by starting to use cork stoppers instead of the traditional wooden stoppers (wood wrapped in hemp soaked in olive oil). With that decision, the wine industry was never the same, with cork stoppers still being used in wine closures [2]. In recent times, natural cork stoppers are used in all types of wines, from centenary to the most recent ones [3].

For instance, the Port wine industry developed faster when it started to age wines in bottles sealed with cork stoppers. This natural material adapts itself correctly to the bottleneck, filling the irregularities and closing perfectly even if the glass expands or contracts, which can happen with a change in the temperature of storage or during transportation. For a long time, cork was claimed to be an inert material as a natural product; however, cork’s chemical inertness has been called into question due to the number of potential compounds that can be extracted [4,5,6,7,8].

There is an interest in investigating wine closures in detail (different types of corks are discussed later in Section 2), including the relationship between the closure’s permeability and the chemical reactions that can occur in wines during bottle storage ageing. Ultimately, such research would be crucial to obtaining a predictive tool to improve wine production and quality [9]. Several reviews have already focused on the wine’s evolution and the main characteristics of cork [9,10,11,12,13,14]. However, none of them showed the correlation that may exist between the wine ageing under a natural cork stopper and the compounds that can be extracted from cork. Thus, the main purpose of this review is to discuss the impact of cork stoppers on wine’s physical-chemical and sensorial properties during ageing in bottles.

Cork from the cork oak, *Quercus suber*, has been substantially studied, mostly because of its economic impact and the worldwide utilisation of cork products [15]. The main commercial product is the natural cork stopper. According to the statistical report of the International Organisation of Vine and Wine (OIV), the annual world wine production is around 275 million hectoliters (292 MhL in 2018), and 70% of the bottled wines are closed with cork stoppers [16].

When the cork is in direct contact with an alcoholic solution such as bottled wine, some cork components can be extracted into the wine [6,8,17]. Hydrocarbons, alcohols, ketones, phenolic compounds including tannins, and other soluble in ethanol/water compounds have been shown to pass from cork to wine and may have a putative contribution to the wine sensory properties of colour, flavour, astringency, and bitterness [18,19,20,21].

### 1.1. Cork Structure

Cork is a natural cellular material with a combination of properties that make it perfect to be used as a wine closure. Hooke [22] first studied the cellular structure of cork, followed by Brugnatelli [23], who discovered that cork was composed of aliphatic, terpenic, and phenolic compounds. These compounds can be obtained by traditional methods or green technologies [24].

Pereira et al. [25] described the chemical composition of cork from *Quercus suber* L. as 0.7% ash, 15.3% total extractives, 38.6% suberin, 21.7% lignin, and 18.2% polysaccharides. The carbohydrate composition analysis showed that glucose represents 50.6% of all monosaccharides, xylose 35.0%, arabinose 7.0%, and galactose and mannose, respectively, 3.6% and 3.4% [25,26]. Furthermore, they described cork as a foam with closed cells formed by the phellogen, with cell division capability (meristematic layer) that produces the bark periderm. The biological tissue is compact with a regular honeycomb arrangement (Figure 1).

The tissue is homogeneous, and cells are constituted by dead parenchymatous cells with hollow, air-filled interiors. The characteristics of the two main chemical components (suberin and lignin, which represent 53% and 26%, respectively, of the cell wall) are responsible for the unique characteristics of cork: impermeability to fluids, low density, buoyancy, low thermal coefficient, high elasticity and deformation behaviour without fracturing if compressed, and long durability [5]. However, cork composition varies with geographic origin, climate, soil conditions, genetic origin, tree dimensions, age (virgin or reproduction), and growth conditions [27,28]. Recently, it has been described that the variability associated with the tree is much more relevant than the effect of the scarcity of water. In practical terms, the climatic changes seem to have a low impact on the performance of cork as a closure for wine bottles [29].

### 1.2. Cork Stoppers Production and Classification

Portugal, Spain, and Morocco are the main suppliers of cork around the world. The cork industry is divided into two main areas: (i) the natural cork industry with natural cork stoppers and natural cork discs; and (ii) the granulate/agglomerated industry that includes technical stoppers, floors, wall coverings, and boards. Natural cork accounts for 70% of the value added by the whole industry. However, more than 70% of the raw material used to make natural cork stoppers is converted into industrial byproducts used in the second sector (granulate-agglomerate industry), improving by this way the total use of the raw material [30].

During bottling, four types of cork can be used: (i) natural cork; (ii) colmated cork; (iii) agglomerated; and iv) technical closures (Figure 2).

The natural cork stoppers are obtained by the direct cutting off from planks of cork wood obtained from cork oak trees. However, despite being only one piece, they can be graded in different categories. The external surface homogeneity/heterogeneity of the natural cork stoppers will determine the commercial grade and the quality classes [31]. The homogeneity of the cork surface is given by the absence of voids or defects (the presence of lenticular channels), which is recognised as the porosity of the cork [32].

The commercial grade and quality classes may go up to nine grades, such as “flor”, which is less porous and the most homogenous, “extra”, “superior”, and “1st to 6th”, with the last ones presenting a higher porosity and more heterogeneity (Figure 3). However, a three-class system, i.e., premium, good, and standard, is a more realistic way to meet the performance requirements [33].

Colmated cork stoppers are made from natural cork where lenticels have been covered with cork powder and glue [35]. The origin of cork powder is from the manufacturing process of improving the performance of the closure since the size of the cork particles and the presence of micro-spheres or other additives in the composition of stoppers are variables that could affect their mechanical behaviour [36]. The agglomerated cork stoppers are made with granulated cork of different dimensions (offcuts of cork from the punching process) and a food-grade binder. This food-grade binder is particularly relevant in the context of this review because, at least in theory, migrations can not only take place from cork to wine but also be derived from the use of these synthetic products. Studies of the potential migration of adhesives and surfaces were made, and a methodology for primary aromatic amines was described as a potential migrant that might be associated with polyurethane adhesives [37]. Another study described 12 neo-formed compounds (amines, amides, and urethane) as a result of the reaction of isocyanates with acetic acid and ethanol used as food simulants, and the levels formed were found to exceed the specific migration limit established [38]. The inorganic elements potentially migrating from cork to a food simulant (a hydroalcoholic solution containing 12 and 20% ethanol) were tested. In all cases, cork met the general safety criteria applicable to food contact materials [39]. In order to avoid direct contact between wine and glue, the technical cork stoppers are constituted of cork granules in the body and natural cork discs on the tops (Figure 2). In this solution, the stoppers maintained the good characteristics of agglomerated closures, and in this case, the wine was in contact with natural cork.

Moreover, synthetic stoppers or plastic have been commercialised since the end of the 20th century and with the main proposal to guarantee wines without “cork taint”. They are designed to look like natural cork and are produced as compressed plastic. For that reason, it is much cheaper than natural cork. Currently, this type represents a negative point because it is made of non-biodegradable material [40]. They can be obtained by two techniques: extrusion and injection moulding, using polymers that are usually used for food packaging. Generally, low density polyethylene (LDPE) for co-extruded stoppers and styrene-butadiene-styrene (SBS) or styrene-ethylene-butylene-styrene (SEBS) for moulded ones [41].

The screw cap is a metal cap that screws onto threads on the neck of a bottle, applying a metal skirt down the neck to become more like the traditional wine capsule. Created in the 1970s, it was the first option to prevent the “cork taint” problem and to reduce oxygen permeation, especially for wines sensitive to oxidation. They are made of aluminium with an inner joint to ensure the barrier property. Two joints are mainly used: the Saran joint, composed of consecutive layers of expanded polyethylene (PEE), polyethylene (PE), tin, and polyvinylidene chloride (PVDC); and the Saranex joint, composed of PEE inserted between two PE/PVDC/PE multilayers [42]. All these components have different proposes: PVDC is used for its good barrier property to oxygen, PE as a water vapour barrier, and PEE for the mechanical property of the joint. The tin layer ensures good sealing of the system.

Finally, the crown capsules are used during the bottle fermentation and ageing of sparkling wines. This type is used as a barrier property to gas transfer in the gasket composed of an LDPE/ethylene vinyl acetate (EVA) blend [43]. Altogether, per year, it was estimated to be around 17 billion closures sold [44]. Take note that 86% of global consumers prefer cork stoppers and, in their opinion, screw caps as well as synthetic closures are dedicated to inexpensive wine [45].

Knowing that cork represents a natural material that can be in direct contact with wine for years, it is of great importance to understand the impact of cork compounds on wine sensory properties. Azevedo et al. [8] quantified the phenolic compounds that were able to be extracted from different classes of cork stoppers (“Flor”, “Third”, and micro-agglomerated), with and without treatment, to wine model solutions. It was observed that natural corks (3rd quality and Flor) were the ones that, at the end of 27 months, allowed a higher quantity of phenolic compounds to be extracted into wine mode solution when compared to agglomerated corks without surface treatment [8]. These results showed that the wine model solution bottled with 3rd quality cork stoppers presented the highest levels of phenolic compounds, which may be related to the highest level of porosity/wooden parts of these stoppers. So as to clarify how porosity influences the profile of volatile organic compounds, a methodology was developed and applied to natural cork stoppers with different levels of porosity: group 1 (low porosity), group 2 (intermediate porosity), and group 3 (high porosity). The differences were found between cork stoppers of low and intermediate porosity when compared with those of high porosity (group 1 vs. 3 and group 2 vs. 3) [46]. The intermediate and low porosity presented higher levels of volatile compounds when compared with group 3. Furthermore, 2-pentylfuran, cyclene, camphene, limonene, eucalyptol, camphor, furfural, and 5-methyl-2-furfural were compounds responsible for defining the highest level of homogeneity within each group, which made possible the subgroups’ creation.

Another goal was the development and validation of a Headspace solid-phase microextraction (HS-SPME) coupled to the gas chromatography-mass spectrometry (GC-MS) method for the quantification of compounds able to differentiate subgroups within each porosity level in wine model solution. This method constitutes a proof-of-concept tool to fine-tune the current selection of stoppers in the cork industry [46]. The primary goal of all of these studies is to identify and quantify the compounds that pass from cork into wine. The main compounds found within the different phenolic classes were phenolic acids and aldehydes, namely gallic and protocatechuic acid amounts of around 3.5 mg/L, and the aldehydes vanillin and protocatechuic quantified in 2.5 and 1.5 mg/L after 27 months of bottling, respectively. More complex polyphenols, namely hydrolysable tannins such as castalagin/vescalagin and mongolícain A/B, were also detected in trace amounts [8]. However, it is possible that hydrolysable tannins could be underestimated and unidentified since this study was focused on low molecular weight polyphenols. According to Reis et al. [47], hydrolysable ellagitannins are present in cork in the same amounts as low molecular weight phenolic compounds [47].

### 1.3. Cork Oxygen Permeability

In the middle of the 19th century, Pasteur was the first scientist to consider the importance of oxygen for wine production and ageing. He wrote: “There exists a period…during which the wine must pass from a permeable container [the barrel] to one nearly impermeable [the bottle]” [48].

The micro-permeability of cork stoppers to oxygen is considered to be the most important property that influences wine sensorial properties during post-bottling. Lopes et al. [49] described that oxygen diffusion varied with the type of closure and material used [49]. The oxygen input through different stoppers was determined by the closure type and was independent of bottle storage position for the generality of the closures tested (during the first 24 months). The oxygen entry rates into bottles were lower in screw caps and “technical” corks, moderate in natural cork stoppers, and higher in synthetic closures [50]. In natural corks, during the first 12 months, oxygen desorbs slowly and continuously from the cell structure into the bottles during storage. In addition, technical cork stoppers are shown to be impermeable to oxygen from the atmosphere during the first 24 months of storage. On the other hand, synthetic closures were permeable to oxygen, mainly after the first month of storage [51]. It seems that the screwcaps are the most airtight systems, followed by technical stoppers, natural corks, and synthetic stoppers, in line with previous reports [11] (Figure 4).

Furthermore, the quantity of oxygen in contact with wine can cause oxidation, which changes the wine aroma [9]. In a study correlating the impact of oxygen in a bottled Bordeaux Sauvignon Blanc wine with commercially available closures, it was observed that wines with higher exposition to oxygen at bottling and sealed with a synthetic with a higher permeability to oxygen presented an oxidised aroma, low in volatile compounds and brown in colour when compared to wines sealed with other types of closures. On the other hand, wine bottles sealed with cork stoppers and screw cap Saranex presented very low levels of reduced and oxidised characteristics [52]. However, due to the fact that the quantities of oxygen in cork stoppers are so low, oxygen transfer from cork stoppers cannot fully explain the evolution of wines after bottling [52]. In fact, in the case of red wines, higher amounts of phenolic compounds make wine easier to oxidise, but at the same time, wines with higher amounts (despite oxidation) become more oxygen resistant. In the case of white wines, the impact is usually perceived as it causes detrimental changes in the aroma, colour, and taste.

However, oxygen permeability has been investigated by different authors using a screw cap closure, two different natural corks, a synthetic closure, and a glass ampoule. The Waters group’s work detailed the results of Riesling and a wooded Chardonnay wine over five years. The main findings were that the synthetic closure produced wines that were relatively oxidised in aroma, brown in colour, and low in sulfur dioxide compared to those held under the other closures. The wines sealed with screw caps or in glass ampoules had a characteristic reduced aroma. This study’s wines sealed with natural corks barely had any reduced aroma. In the circumstances of this study, bottle orientation during storage had no discernible effects on the composition and sensory characteristics of the wines under evaluation [53]. Another work concerning the consistency of “alternative” closures was developed by Brotto et al. [54], which described a technique used to assess the variation among the lot of 20 different types of stoppers regarding oxygen permeability. The synthetic closure performed more consistently, while the natural cork stopper displayed low homogeneity within the batch, notably in the first month following bottling [54]. Focus only on natural cork stoppers and the capacity of oxygen ingress capabilities. Oliveira et al. [55] studied 600 natural cork stoppers made from cork boards in order to evaluate various quality classes. They are examined in order to account for the cork’s inherent variability in relation to oxygen intrusion into the bottle. All examples had similar oxygen transport kinetics that could be adapted to logarithmic models. A substantial variation was discovered in the oxygen entry into the bottles sealed with natural cork stoppers. The findings imply that variations in oxygen intrusion are caused by the stopper’s structure’s naturally occurring variations in cell size and air volume [55].

However, technological advances allow the creation of novel types of closures, so the ranking related to oxygen permeability of different closures is modified. More recent work by [56] over a 10-year period examined the chemical alterations in the strength of the oxidation odour in three Sauvignon blanc wines sealed with natural cork and various closures that had varied known oxygen transfer rates. Free SO_2_ and 3-sulfanylhexanol loss during ageing were connected with closure oxygen transfer rate levels, as were increases in dissolved O_2_, OD420, and sotolon. Following a 10-year ageing period, sensory analysis was carried out, accompanied by additional chemical analysis of fragrance effect markers, and it was discovered that some Sauvignon blanc wines were protected from oxidation as long as the closing oxygen transfer rate did not surpass 0.3 mg/year.

It is also worth considering that the high inconsistency of cork oxygen ingress has been reported to cause variable levels of dissolved oxygen in the wine during bottle storage, which in turn has been associated with variable levels of sotolon, a well-known oxidation marker [57]. Seven years after bottling, the sotolon and oxygen contents of various bottles of the same white wine were studied. The sotolon content in wine continued to be below its perceptibility threshold at the range of dissolved oxygen concentrations typically observed. Only in wines with oxygen contents exceeding 500 g/L was the perceptual threshold surpassed. The amount of free sulphur dioxide in the wine samples under study decreased as a result of the presence of dissolved oxygen [58].

Therefore, the impact of oxygen on bottled wine is going to differ depending on the type of wine [59]. Moreover, since the 1960s, industry and scientists have been working together to identify compounds present in wines, especially phenolic compounds, involved in the mechanisms of oxidation occurring in wines [9].

### 1.4. Phenolic Compounds in Cork

Since 1990, several studies have characterized the chemical composition of cork, mainly on phenolic compounds (Table 1) [6,7,8,18,60,61,62,63]. The first compounds that were identified in cork samples were gallic, sinapic, caffeic acids, and vanillin [6,62,64]. Ten years later, the ellagitannins vescalagin and castalagin (isomers), grandinin, and roburin were reported as the main compounds that were extracted from a wine model solution after 24 h of contact with cork stoppers [17]. Furthermore, the structures of 33 compounds present in cork from *Quercus suber* L. were tentatively identified by LC-MS after being extracted into hydroalcoholic solutions (Table 1) [7].

The simpler phenolic compounds detected were phenolic acids and aldehydes and more complex such as gallic acid derivatives, gallotannins (galloyl esters of glucose), combinations of galloyl and ellagitannins (hexahydroxydiphenoyl esters of glucose), dehydrated tergallic-C-glucosides, or ellagic acid derivatives. In addition, Mongolícain (flavanoellagitannin with hydrolysable tannin and flavan-3-ol moieties are linked through a carbon-carbon bond) was also detected in cork from *Quercus suber* L. [66]. These examples show how rich in phenolic compounds cork is [7]. Santos et al. [62] reported for the first time salicylic acid, eriodictyol, naringenin, quinic acid, and hydroxyphenyllactic acid as cork components [62]. Moreover, Reis et al. [47] described the presence of vescalin, castalin, guajavin B/eugenigrandinin A, vescavaloninic, and castavaloninic acids in cork samples [47]. Other ellagitannins such as the glycosylated structure of acutissimin A/B and guajavin B/eugenigrandinin A, oligomeric ellagitannins and a glycosylated dimer of gallocatechin linked to vescalagin/castalagin were found to occur by MALDI-TOF [47]. In another work, a wine model solution extracted ellagitannins, proanthocyanidins, and pectic-derived polysaccharides from natural cork stoppers, with simple C-glycosidic, complex, and oligomeric ellagitannins being identified by HPLC-DAD/ESI-MS. In addition, MALDI-TOF-MS was used for the identification of ellagitannins linked to proanthocyanidins and some pectic-derived polysaccharides [68].

Some studies were developed to evaluate the potential correlation between the composition of cork and its geographic region of origin. Cork stopper granulates from 11 geographical origins in Portugal and Spain were analysed by HPLC-DAD/ESI-MS and near-infrared spectroscopy (NIRS) to assess geographical discrimination regarding polyphenol composition. NIRS technique showed to be a powerful tool to discriminate origins and predict the concentration of polyphenols. However, the variability in the phenolic compound composition of cork samples was high, and it was not influenced by geographical location [27]. Within the same regions, Guedes de Pinho et al. [28] demonstrated three main clusters of regions according to their chemical similarity; however, geographical proximity was not found [28]. Nineteen compounds were responsible for the clusters, including terpenes, polyphenols (vescalagin and castalagin), and others (pyrogallol, glucosan, sitost-4-en-3-one, *o*-cymene, quinic acid, and five unidentified compounds), with the authors concluding that the geographical location does not seem to be responsible for the variability in the polyphenol composition of cork stoppers, being more likely influenced by genetics or tree age [28]. The fact that there are no statistical differences between each of the regions studied may be due to the high amounts of gallic, protocatechuic, and ellagic acids present in samples from all regions [60,64].

### 1.5. Interaction between Cork and Wine

Cork stoppers are known to be the major cause of contamination in wines [69]. Cork taint is, in fact, a problem in this industry, and a recent review described in detail six compounds that have been found to contribute to this undesirable flavour. These are guaiacol, geosmin, 2-methylisoborneol (MIB), octen-3-ol, and octen-3-one; and the most important of them all, 2,4,6 trichloroanisole [10]. Other authors also described 2-Methoxy-3,5-dimethylpyrazine [70,71]. The geosmin and 2-methylisoborneol (2-MIB) are responsible for earthy off-flavour; pyrazines cause vegetable odours, and guaiacol results in smoked, phenolic, and medicinal defects; and the 1-octen-3-ol and 1-octen-3-one cause off-odours of mushrooms in wines which are caused by grapes contaminated by bunch rot [10]. Moreover, the best known and the most problematic off-flavour compound of the cork stopper is the 2,4,6-tricloroanisole (TCA), known specifically to yield a “cork taint” in wines [72], and it is formed from the chlorination of lignin-related compounds during chlorine bleaching in the processing of cork [73]. About 20 years ago, this problem in the wine industry originated losses of around 10 billion dollars per year [74]. The small amounts around ng/L of TCA can be extremely detrimental to the wine quality [75], derived from its high volatility and low perception threshold (1.5–2 ng/L) [69]. Other compounds from the same family can also be responsible for the designated cork taint, but in a much less extension, the 2,4-dichloroanisole (2,4-DCA), 2,6-dichloroanisole (2,6-DCA), 2,3,4,6-tetrachloroanisole (TeCA), and pentachloroanisole (PCA). Furthermore, the 2,4,6-tribromoanisole (TBA) has a significant impact on the musty/mouldy fault; however, this compound is not found in cork [76]. The chlorophenols (2,4,6-trichlorophenol, 2,3,4,6-tetrachlorophenol, and pentachlorophenol (PCP)) were also described as “cork taint responsible compounds” [72]. Note that, not always, the sources of contamination are natural cork stoppers. The air pollution of the cellars, wood materials, barrels, and chips are also known as the sources of the problem [10].

In addition, several groups have been working on the identification of the main volatile compounds present in cork, such as aliphatic alcohols, monoterpenes, triterpenes, sterols, phenols, and fatty acids [77,78], described as the main compounds (Table 2). These compounds can be extracted from cork by a wine model solution in a bottle [28].

In addition to the volatile fraction, polyphenols can be extracted from cork to wine model solutions [8,66]. Therefore, it is important to understand the possible impact these compounds may have on the evolution of wine sealed with cork stoppers. A recent study showed that corks from different origins (suppliers) could impact differently on wine properties [79] due to the phenolic acids susceptible to passing from cork to wine. Gallic, caftaric, caffeic, and *p*-coumaric acids were studied in two-disc corks from three different commercial suppliers. Gallic acid was significantly higher in Cork A wines, which indicates the contribution of Cork A to the concentration of this compound in the wine. This may be due to differences in the surface roughness of cork that would increase the surface area in contact with the wine [79]. Moreover, a study evaluated the reactivity of phenolic compounds from cork stoppers and assayed their reaction with two major wine components, namely, (+)-catechin, and malvidin-3-*O*-glucoside. From these assays, several compounds already described in the literature were detected together with a new compound belonging to a family of ellagitannin-derived compounds named corklins (Figure 5) [67]. The interaction between ellagitannins in alcoholic solutions and catechins (cat) described this new family of compounds.

Other reactions yielded the formation of structures such as pinotins (Figure 5) [80], xanthylium salts [81], and a dimer of cat-Vanillin-cat [82]. All these compounds have different chromatic characteristics from their precursors and can thus have an impact on the final colour of wines. Indeed, phenolic acids and aldehydes have been identified as the major compounds that pass through to bottled wine model solutions from various cork stopper grades [8]. These phenolic acids did not directly influence the organoleptic properties of red wine but may be involved in white wine colour evolution when oxidised to the respective quinones [9]. On the other hand, a recent work describes the influence that these compounds may have on the sensory perception of wine from an astringency perspective [18].

In wine, tannins can undergo chemical transformations such as condensation with anthocyanins, pyranoanthocyanins, or other phenolic compounds, oxidation, or hydrolysis reactions in a slow but continuous way [83], yielding to the formation of more complex structures such as portisins [84], flavano-ellagitannins, and ethyl-vescalagin [85] or corklins (ellagitannin derivate compounds) [68]. While it is known that grape seed tannins contribute to the structure of wines, they can also cause excessive astringency. When combined with polysaccharides and proteins, tannins contribute to wine’s softness and roundness but can also impart herbaceous notes if grapes are not ripened [86]. These compounds have also been described as some of the compounds that pass from cork stopper to model wine solutions [17,66].

From an astringency standpoint, a recent work [18] explored for the first time the interaction between three cork fractions and salivary proteins. The results suggested that the ellagitannin castalagin was the most reactive with salivary proteins, while 4-dehydrocastalagin was the most precipitated compound. The fraction containing caffeic and sinapic acids showed the highest interaction with salivary proteins, mainly cystatins. Castalagin, 4-dehydrocastalagin, caffeic, and ellagic acids were almost always precipitated above their reported astringency thresholds, and by this, they may contribute to the astringency perception.

The wine’s flavour undergoes changes during bottle ageing, which can be induced by the type of closure (cork, synthetic, or screw cap). These changes are more noticeable when the storage period is longer than the five years described by Guedes de Pinho and co-workers [12]. The permeation, sorption (scalping), or desorption phenomena among closures and wines were studied and concluded that corks helped to keep the aroma, as cork material had a low sorption capacity for some compounds, specifically non-polar ones, which were known to be directly correlated with fruity wine scents. [12]. Moreover, the authors also reported for the first time the presence of two new compounds, *trans*-4-*tert*-butylcyclohexanol and *2,4*-di-*tert*-butylphenol, in wines sealed with specific micro-agglomerated cork stoppers with plastic microspheres and synthetic closures, respectively. Furthermore, wines with synthetic stoppers presented high levels of oxidative attributes, low levels of SO_2_, and high colour intensity, contrarily to cork stoppers that demonstrated higher scores in aroma intensity, aroma quality, and balance [87].

As previously mentioned, cork stoppers as well as wood barrels could impact wine organoleptic properties (aroma, colour, and taste) during ageing. Despite extensive knowledge on the subject, oenologists typically select cork stoppers or wood barrel varieties based on intuition rather than targeted chemical interactions. A recent study suggested that some polyphenols found in cork stoppers migrate to wine solutions, interacting with human salivary proteins linked to astringency. This could be positive or negative, depending on the type of wine. This knowledge should be expanded to other varieties of cork stoppers in order to allow the choice of stoppers to upgrade wine quality during storage and ageing. Currently, numerous studies are trying to correlate the beneficial effect of cork on wine’s physical-chemical and sensorial parameters.

## 2. Sustainability

Cork oak trees are natural retainers of CO_2_, with a stripped cork oak absorbing, on average, five times more CO_2_ during the regeneration than a regular cork oak [88]. Moreover, these trees prevent soil degradation, regulate the hydrological cycle, combat desertification, and contribute to biodiversity. Cork is a natural material that is both renewable and recyclable [89]. Cork is not only used in the wine industry, and its application in other industries has been increasing, such as in footwear, automotive, and packaging industries. In the cork industry, there is no waste, and everything is transformed into new products and solutions [2].

The cork industry’s main byproducts are cork powder and granules. This product has a wide range of applications, from cork stoppers to incorporation in agglomerates and briquettes to use as an adsorbent in the treatment of gaseous emissions, waters, and wastewaters [90]. In its natural form, cork biomass has been used as a biosorbent for heavy metals and oils, and it is also a precursor of activated carbon for the removal of emerging organic pollutants in water. [91].

Several research groups are looking for non-destructive methods to remove some undesirable volatile compounds from cork that could eventually end up contaminating bottled wines [13,92]. These techniques have been developed with the purpose of being a clean, sustainable, and efficient process [93]. Supercritical fluids extraction in the cork industry has been applied to extract contaminants such as TCA or bioactive compounds [74,92,94]. This methodology aims to produce clean cork granules, ensuring that microagglomerate cork is free of TCA and contaminants.

This knowledge is very important within the context of the circular economy, as the characterisation of natural waste sources for reutilisation leads to greater global sustainability [95]. In general, cork stoppers and cork byproduct extracts demonstrated high aromatic and antioxidant potential to be further reused in different industries, such as agricultural, cosmetic, and pharmaceutical industries [8,16,47,60,83,92,96].

In a similar vein, the work described by the Freitas group [97] uses natural deep eutectic solvents (NADES) with the purpose of eliminating or reducing the use of toxic chemicals. They used different natural compounds, such as lactic acid and glycerol. The results showed higher extraction capacity with acidic NADES, which extracted great quantities of aromatic compounds, terpenoids, and fatty acids. On the other hand, more basic eutectic mixtures extracted more quantities of low molecular weight polar compounds. This research, along with the associated nontoxicity, low cost, and ease of preparation, establishes NADES as a green approach to extracting high-added-value compounds from cork [97].

When considering adsorption for water treatment, commercial activated carbon is typically the adsorbent of choice for the removal of pollutants from the aqueous phase, particularly pharmaceuticals. Attempts have been made to use wastes as raw materials for the production of alternative carbon adsorbents in order to reduce costs and conserve natural resources. This approach aims to improve efficiency and cost-effectiveness while also proposing an alternative and sustainable method for residue valourisation and management [90,98]. Cork has gained popularity in water remediation since it was first used as a precursor for the production of eco-friendly activated carbon through chemical and physical activation. The findings of this study show that lab-made carbons have adequate properties for removing pharmaceutical compounds from water [99]. Furthermore, cork powder biosorption is regarded as a promising method for heavy metal removal from industrial wastewaters, such as those from chromium tanning factories [100].

In a circular economy approach, it is intended to close the energetic cycle. Despite the technical and safety difficulties presented by the use of cork powder as a low-density material, which complicates its transportation for industrial uses outside the area in which it is produced, the cork industry has attempted to take advantage of residues, primarily through direct energy recovery. As a result, cork pellets emerge as a safer and more easily transportable alternative for energy recovery from cork dust and other granulated types of cork waste, with the potential for wider application. These findings show that cork pellets have a higher calorific value than other biomass pellets [101].

## 3. Conclusions

As a result, the recognition of cork as an excellent product used globally has piqued the interest of this research area. The heterogeneity of its chemical constitution and its remarkable properties give cork material enormous potential and considerable importance. The wine industry and the scientific community have published a number of sensory studies that have reported that consumers believe that wines bottled with cork stoppers present an enhanced quality compared to wines sealed with other closures [102].

There is also a valourisation as a natural product from a sustainable perspective and also an accounting that its byproducts can be regarded as a relevant source of bioactive components, such as phenolic acids, terpenoids, and tannins, as already described in the literature [47,66,92,96].

The path of research seems to point to the cork stopper as an excellent closure. It is a fact for the cork industry that the ability to predict whether the cork may or may not influence bottled wine will always be an asset. Ultimately, the cork stopper will always be a natural choice.

## Figures and Tables

**Figure 1 foods-11-02638-f001:**
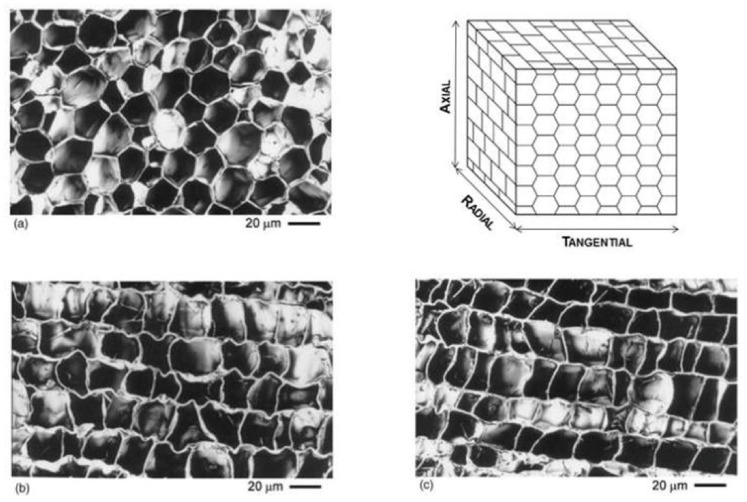
Schematic representation of the cellular structure of cork with scanning electron micrographs of sections of reproduction cork: (**a**) tangential, (**b**) radial, and (**c**) transverse sections, from [14].

**Figure 2 foods-11-02638-f002:**
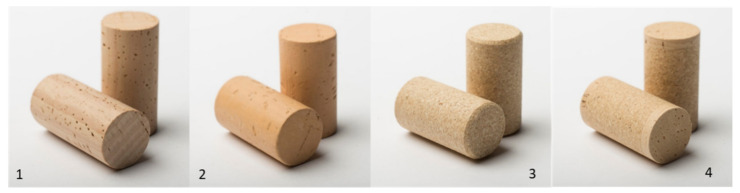
Types of cork closures that can be used for wine bottling (https://www.corklink.com/index.php/natural-wine-cork-classification/ accessed on 18 July 2021). (**1**)—corresponding to natural cork stoppers; (**2**)—Colmated cork stoppers; (**3**,**4**) corresponding to technical cork stoppers: closures made by cork granulate (agglomerated).

**Figure 3 foods-11-02638-f003:**
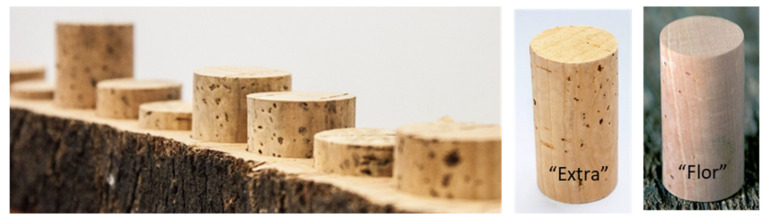
Natural cork stoppers (https://www.corklink.com/index.php/natural-wine-cork-classification/ accessed on accessed on 24 May 2021). [34].

**Figure 4 foods-11-02638-f004:**
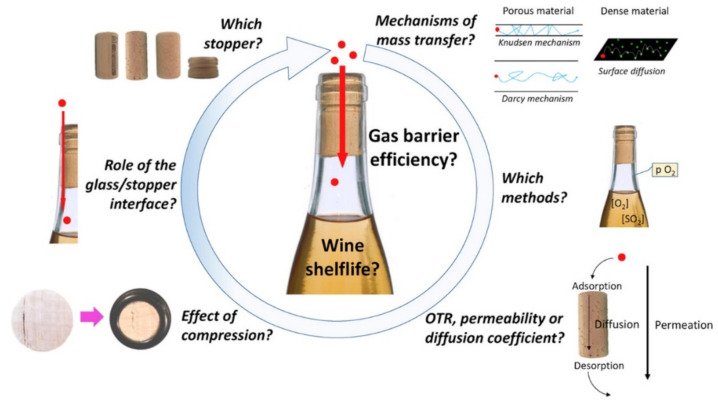
Gas transfer through wine closures (graphical abstract from [11]).

**Figure 5 foods-11-02638-f005:**
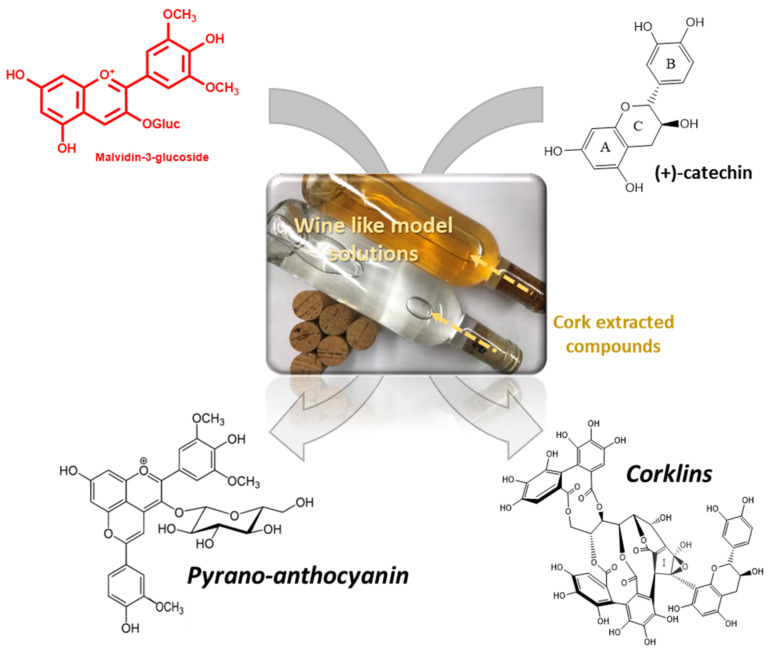
Bottling study scheme and compounds formed (graphical abstract from [69]).

**Table 1 foods-11-02638-t001:** Compounds found in cork.

Compound	First Described in *Quercus suber*
Gallic acid	[6]
Protocatechuic acid	[6]
Protocatechuic aldehyde	[6]
Coniferaldehyde	[6]
Caffeic acid	[6]
Ferulic acid	[6]
Vanillin	[6]
Sinapic acid	[6]
Ellagic acid	[6]
Ellagic acid-pentose	[61]
Ellagic acid-deoxyhexose	[6,61]
Ellagic acid-hexose	[61]
Valoneic acid dilactone	[6,61]
HHDP*-glucose	[65]
Valoneic acid	[65]
Dehydrated tergallic-C-glucoside	[65]
HHDP*-galloyl-glucose	[65]
Trigalloy-glucose	[65]
Di-HHDP*-glucose	[65]
HHDP*-digalloyl-glucose	[65]
Tetragalloyl-glucose	[65]
Di-HHDP*-galloyl-glucose	[65]
Trigalloyl-HHDP*-glucose	[65]
Pentagalloyl-glucose	[65]
Mongolicain	[66]
Dehydrocastalagin	[18]
Ellagic acid rhamnoside	[18]
Vescalagin	[17]
Castalagin	[17]
Vescalagin-ethanol derivative	[67]
Roburin A	[17,61]
Roburin E	[61]
Granidin	[17]
Ethyl-vescalagin	[67]
Acutissimin	[67]
salicylic acid	[62]
erioctyol	[62]
naringenin	[62]
quinic acid	[62]
hydroxyphenyllactic acid	[62]
vescalin	[47]
castalin	[47]
guajavin B/eugenigrandinin A	[47]
vescavaloninic acid	[47]
castavaloninic acid	[47]
Isorhamnetin-3-O-rutinoside	[18]

* HHDP: hexahydriphenyl.

**Table 2 foods-11-02638-t002:** Description of semi-volatile (GC-MS) and volatile (HS-SPME-GC-MS) compounds able to pass from cork by methanol and wine model solution [28].

**GC-MS Profiling of Semi-Volatile Compounds Extracted from Cork by Methanol**				
*Alkenes*	*Carbohydrate*	*Fatty alcohols*		
1-Pentacosene	Glucosan	1-Eicosanol		
1-Heptacosene		1-Docosanol		
*Fatty acids*		*Glycerolipids*		
n-Hexadecanoic acid		Glycerol 1-hexadecanoate		
cis-9,cis-12-Octadecadienoic acid		Glycerol 1-octadecanoate		
cis-9-Octadecenoic acid				
n-Octadecanoic acid				
Eicosanoic acid				
Docosanoic acid				
*Phenols and derivatives*	*Sterols*		*Triterpenes*	
Catechol	Stigmastan-3,5-diene		*trans*-Squalene	
Pyrogallol	β-Sitosterol		Lupen-3-one	
Vanillin	Sitost-4-en-3-one		Lupeol	
*trans*-Coniferyl alcohol			Friedelin	
			Betulin	
**HS-SPME-GC-MS Profiling of Volatile Compounds Extracted from Cork by Wine Model Solution**				
*Aldehydes*	*Benzenoids*	*Esters*		*Monoterpenes*
Hexanal	o-Cymene	Ethyl hexanoate		α-Pinene
Heptanal	Naphthalene	Ethyl heptanoate		Camphene
Benzaldehyde		Ethyl nonanoate		β-Pinene
Octanal		Fenchyl acetate		1,4-Cineole
Nonanal		Isobornyl acetate		α-Terpinene
Decanal				Limonene
				Eucalyptol
*Sesquiterpenes*				Terpinolene
α-Copaene				Fenchone
d-Longifolene				Fenchol
β-Cadinene				α-Campholenal
l-Calamenene				(+)-Camphor
Eremophila ketone				trans-β-Terpineol
Sesquiterpene 1				trans-3-Pinanone
Sesquiterpene 2				Isoborneol
Sesquiterpene 3				l-Borneol
Sesquiterpene 4				cis-3-Pinanone
Sesquiterpene 5				1-Terpinen-4-ol
				α-Terpineol
				Monoterpene 1

## Data Availability

The data that support the findings of this study are available from the corresponding author upon reasonable request.

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
