# Peer review of "Cork, a Natural Choice to Wine?"

_foods, 2022, doi:10.3390/foods11172638_

Round 1

Reviewer 1 Report

Dear Authors, I have reviewed the article entitled: Cork, a natural choice to wine?

I suggest the following observations:

- Please revise the title because it is not clear that the study represents an analysis of specialized scientific literature.

-The abstract must be modified so that the importance of the study and the main purpose are highlighted. It should be clear what will be found in the work and by what methods the analysis will be done.

-Remove “Several aspects will be reviewed” from line 10.

Line 215. Why a figure was called graphic abstract introduced in the work. Either rename the figure, or attach it as an additional document.

-The manuscript does not provide any cutting edge information, however it has a lot of work and it can be a nice compilation of studies done in the subject. However the organization of the manuscript should be changed and improved.

-After reorganizing the material, I suggest creating a table of contents that will be presented at the beginning of the work so that the content can be followed more easily.

-There are several essential books on this field, which have the same fundamental information, but which are not found in the bibliography. I believe that they should be taken into account, considering that they were the basis of the study of the use of cork stoppers.

-I think that more attention should be paid to the other ways of closing the wine in the bottle. Only after discussing these possibilities can relevant conclusions be drawn. Otherwise, a conflict of interest may arise.

Author Response

- Please revise the title because it is not clear that the study represents an analysis of specialized scientific literature.

Response: We agree to put the title this way to arouse interest in reading the article, being different, try to show that the article will describe the various aspects of cork when used to close bottles

-The abstract must be modified so that the importance of the study and the main purpose are highlighted. It should be clear what will be found in the work and by what methods the analysis will be done.

Response: From our point of view, being a compilation of several works carried out, we do not think that methods and specifications are described in this field. They are described throughout the text and as they are relevant.

-Remove “Several aspects will be reviewed” from line 10.

Response: It was removed and sentence reformulated.

Line 215. Why a figure was called graphic abstract introduced in the work. Either rename the figure, or attach it as an additional document.

Response: The title was changed to “Gas transfer through wine closures.”

-The manuscript does not provide any cutting edge information, however it has a lot of work and it can be a nice compilation of studies done in the subject. However the organization of the manuscript should be changed and improved.

Response: After several changes, all the authors of the article agreed that the way it was structured, and presented here, would be the easiest and most appealing way of reading and interconnecting the contents discussed. I hope that with the increase in the table of contents placed, you will be able to agree with us.

-After reorganizing the material, I suggest creating a table of contents that will be presented at the beginning of the work so that the content can be followed more easily.

Response: The table of contents was added. It was added after the abstract, I don´t saw information where put this information in FOODS template.

-There are several essential books on this field, which have the same fundamental information, but which are not found in the bibliography. I believe that they should be taken into account, considering that they were the basis of the study of the use of cork stoppers.

Response: More relevant literature were added.

-I think that more attention should be paid to the other ways of closing the wine in the bottle. Only after discussing these possibilities can relevant conclusions be drawn. Otherwise, a conflict of interest may arise.

Response: I understand what you mean, but we focus on the interest and theme of cork. Other types of stoppers are described in other review articles.

Reviewer 2 Report

The manuscript foods-1865976 entitled "Cork, a natural choice to wine?" is clearly organized and the subject matter is suitable. The paper brings us recent data to understand the topic of cork in wines. 

Title and abstract: Titles structured as questions are good choices to attract the reader's initial attention, inviting them to read the abstract. The title of this study provokes the reader, asking whether cork is the natural option for wine products. The abstract is well-written with a selection of the most important information that could be found in this review. Moreover, the review contains a significant number of references. However, these references are not prepared according to the author's guidelines. I don't have any major suggestions, but only some minor comments.

L. 35 delete "."

L. 63, 162, 193, 200 should be e.g. Pereira et al. [20], Azevedo et al. [8], Reis et al. [43] etc.

L. 67 please re-write the phrase "with closed cells with cells being formed".

L. 180 HS-SPME-GC-MS/MS: full name of this method has to be added

L. 293 delete year, should be "Santos et al. [66]", the same L. 295 "Reis et al. [43]"

L. 366 "-O-" should be written in italics

L. 372 add ")"

Author Response

Response: the endnote style was changed from “Numbered” for “MDPI” style.

  1. 35 delete "."

Response: the point was removed.

  1. 63, 162, 193, 200 should be e.g. Pereira et al. [20], Azevedo et al. [8], Reis et al. [43] etc.

Response: the information was added.

  1. 67 please re-write the phrase "with closed cells with cells being formed".

Response: the phrase was re written.

  1. 180 HS-SPME-GC-MS/MS: full name of this method has to be added

Response: the information was added.

  1. 293 delete year, should be "Santos et al. [66]", the same L. 295 "Reis et al. [43]"

Response: the year was deleted.

  1. 366 "-O-" should be written in italics

Response: it was corrected.

  1. 372 add ")"

Response: it was added.

Reviewer 3 Report

This review summarizes the current knowledge on compounds in cork and attendant influence on the wine quality and delves further on other utilities of cork-derived byproducts

There are issues to be addressed by the authors:

Line 35. Check sentence punctuation

Line 39. Replace the word ‘work’ with ‘research’

Line 40-41. Provide reference.

Line 61. The sentence is incomplete

Line 215. Title of Figure 4 is not descriptively comprehensive

Line 216. The author alleges that oxygen causes reduction is incorrect

Line 327. Use consistent colour for the figures

Line 441. Check grammar

Line 452. The word solvent is repeated. Correct and check grammar

Most of the references are not formatted according to journal style.

Author Response

Line 35. Check sentence punctuation

Response: the point was removed.

Line 39. Replace the word ‘work’ with ‘research’

Response: the word was replaced.

Line 40-41. Provide reference.

Response: the reference was added.

Line 61. The sentence is incomplete

Response: the sentence was re-written.

Line 215. Title of Figure 4 is not descriptively comprehensive

Response: the title was changed.

Line 216. The author alleges that oxygen causes reduction is incorrect

Response: the “reduction” was removed.

Line 327. Use consistent colour for the figures

Response: ok.

Line 441. Check grammar

Response: The grammar was corrected.

Line 452. The word solvent is repeated. Correct and check grammar

Response: it was corrected and checked.

Most of the references are not formatted according to journal style.

Response: the endnote style was changed from “Numbered” for “MDPI” style.